# Clinical anemia predicts dermal parasitism and reservoir infectiousness during progressive visceral leishmaniosis

Max C. Waugh[1,2]*, Karen I. Cyndari[2,3,4], Tom J. Lynch[5], Soomin Koh[1,2,6], Ferney Henao-Ceballos[6], Jacob J. Oleson[6], Paul M. Kaye[7], Christine A. Petersen[1,2]

1 Department of Epidemiology, College of Public Health, University of Iowa, Iowa City, Iowa, United States of America, 2 Center for Emerging Infectious Diseases, University of Iowa, Iowa City, Iowa, United States of America, 3 Department of Emergency Medicine, University of Iowa, Iowa City, Iowa, United States of America, 4 University of Iowa Hospitals and Clinics, Iowa City, Iowa, United States of America, 5 Department of Anatomy and Cell Biology, University of Iowa, University of Iowa, Iowa City, Iowa, United States of America, 6 Department of Biostatistics, College of Public Health, University of Iowa, Iowa City, Iowa, United States of America, 7 York Biomedical Research Institute, Hull York Medical School, University of York, United Kingdom

* waugh.200@osu.edu

**Data Availability Statement:** Data supporting the findings of this study is available on researchgate.net under Christine Anne Petersen's profile at

## Abstract

Dogs represent the primary reservoir for *Leishmania infantum* human visceral leishmaniasis (VL) transmitted through phlebotomine sand flies. Public health initiatives targeting zoonotic VL commonly focus on dogs with severe clinical disease, often in renal failure, as they have previously been considered the most infectious to sand flies. However, more recent studies suggest that dogs with mild to moderate clinical disease may be more infectious than dogs with severe disease. The mechanisms of infectiousness from the skin and how this relates to transmissibility as clinical disease progresses is largely unknown. We evaluated dermal parasitism in dogs naturally infected with *L. infantum* across the four LeishVet clinical stages of disease. We establish the relationship between dermal parasitism, critical, frequently observed, clinical parameters such as anemia and creatinine, and infectiousness. Using RNAscope and confocal microscopy, we found notable variation in dermal parasitism between dogs, particularly within LeishVet II. Dogs with mild disease had significantly less dermal inflammation and parasitism than dogs with moderate or severe disease. We found significant correlations between anemia, dermal parasitism, and infectiousness (p = 0.0098, r = -0.4798; p = 0.0022, r = -0.8364). In contrast, we did not observe significant correlation between creatinine, a measure of renal function, and dermal parasitism or infectiousness. Host blood cell abnormalities, including anemia, correlate with infectiousness to sand flies. As these signs of disease often appear earlier in the course of disease, this indicates that classical measures of disease severity do not necessarily correlate with infectiousness or epidemiological importance. Public health initiatives attempting to break the zoonotic cycle of *L. infantum* infection should therefore focus on preventing transmission from infectious, anemic dogs, and not those with the most severe disease.

https://www.researchgate.net/publication/384733165_Plos_NTD_RNAscope_Data_to_RG_OCt_2024.

**Funding:** This work was funded by R01 AI171971 from the National Institute of Allergy and Infectious Diseases, National Institutes of Health. This work was completed while M.C.W. was supported by the University of Iowa Interdisciplinary Immunology Postdoctoral Training Grant T32 AI007260, National Institute of Allergy and Infectious Diseases, National Institutes of Health. The funders had no role in study design, data collection and analysis, decision to publish, or preparation of the manuscript.

**Competing interests:** The authors have declared that no competing interests exist.

## Author summary

Dogs infected with *Leishmania* can transmit parasites to humans and other dogs through sand flies. This causes human visceral leishmaniasis (VL) and canine leishmaniosis (CanL). Public health interventions removing clinically ill dogs from the population have not significantly decreased rates of human or canine disease. Additionally, studies found that dogs with mild to moderate clinical disease transmitted the most parasites to feeding sand flies. We evaluated skin sections from *L. infantum*-infected dogs and characterized the dermal and systemic factors associated with infectiousness to sand flies. We found that the number of parasitized immune cells positively correlated with infectiousness to sand flies. Additionally, anemia, measured by a decreasing hematocrit, significantly correlated with increased parasitized dermal immune cells and infectiousness to sand flies. In contrast, parasitized dermal immune cells did not significantly change as chronic kidney disease, measured by creatinine, progressed, even though these were the sickest animals based on the LeishVet clinical scoring system. These initial findings suggest that early host cell-parasite interactions underlying clinical parameters, more than overall clinical severity, influences infectiousness to sand flies. Public health initiatives should focus on breaking the cycle of zoonotic infection by providing insecticide intervention to dogs most infectious to sand flies.

*Leishmania infantum*, an obligate intracellular protozoan parasite, causes significant global morbidity and mortality in humans and domestic reservoir, dogs [1,2]. In endemic regions, sand flies transmit the parasite between dogs and humans [1]. Skin is the critical interface of transmission; *Leishmania* amastigotes concentrate there in a heterogenous distribution hypothesized to promote uptake by feeding sand flies [3,4]. Preventing sand fly feeding, such as through insecticide-impregnated collars for dogs, has been shown to significantly decrease *L. Infantum* exposure [5].

*L. infantum*-infected dogs show a variety of clinical signs, which depend on their immune response to the parasite. Subclinical dogs maintain a systemic Th1 type immune response that promotes macrophage activation and parasite destruction [6]. With chronic antigen exposure, this pro-inflammatory response diminishes and shifts to an anti-inflammatory, regulatory response that does not control parasite replication and promotes antibody over-production [6]. Many CanL clinical signs originate from uncontrolled non-neutralizing antibody production, parasite replication, and ineffective immune response [7]. For example, hypergammaglobulinemia promotes antigen-antibody complex formation in the renal glomeruli, causing glomerulonephritis detectable as chronic kidney disease (CKD) [7–10]. CKD is a common clinical finding and a major cause of mortality in CanL [7,10].

Aside from CKD, hematologic pathology, including anemia, thrombocytopenia and epistaxis are frequently reported in clinical CanL [2,7,11]. Multiple processes contribute to this pathology; however, disrupted hematopoiesis due to bone marrow parasitism likely plays an important role [7,11–13]. Parasitism of bone marrow shifts local cytokine production to increase proinflammatory cytokines, which affects hematopoiesis, and alters the histological appearance of the bone marrow [7,13–15]. These bone marrow changes could be creating an environment that promotes the survival and expansion of *Leishmania*. Bone marrow pathology also correlates with shifts in circulating leukocyte and erythrocyte populations, illustrating how local parasitism and pathology have systemic effects [14,16,17].

Previous studies often classified CanL severity based only on physical examination vs. including clinicopathological measures, which complicates evaluation of how clinical disease affects infectiousness [18–22]. CanL clinical severity classification systems have been published, such as the LeishVet staging system, which uses physical exam findings, bloodwork changes, and serology to differentiate four stages of disease severity, primarily based on the severity of CKD [2,23]. Scorza et al used the LeishVet staging system to help standardize their evaluation of the relationships between clinical severity, tissue parasite burden, and infectiousness to sand flies. They found that dogs with a high dermal parasite load were the most infectious to sand flies [24]. Interestingly, these dogs often presented with mild to moderate clinical disease, based on the LeishVet scoring system, suggesting that clinical severity alone does not drive transmission [24]. This study establishes how the dermal immune environment and *L. infantum* burden changes as clinical disease progresses, and how differences in parasite-mediated inflammation affected the dermal immune environment and host infectiousness to sand flies.

## Materials and methods

### Ethics statement

All animal use involved in this work was approved by the University of Iowa Institutional Animal Care and Use Committee and was performed under the supervision of licensed and, where appropriate, board-certified veterinarians according to International AAALAC accreditation standards.

### Animal cohort

The first part of this study was retrospective, using n = 20 archived skin samples from *L. infantum* infected dogs collected between 2011 and 2022, as depicted in the CONSORT flow diagram (Fig 1) [25]. These samples were collected as previously described (23). The second part of this study was prospective (n = 17). All samples were collected with owner/caretaker informed consent from a cohort of U.S. hunting hounds where vertically transmitted *L.*

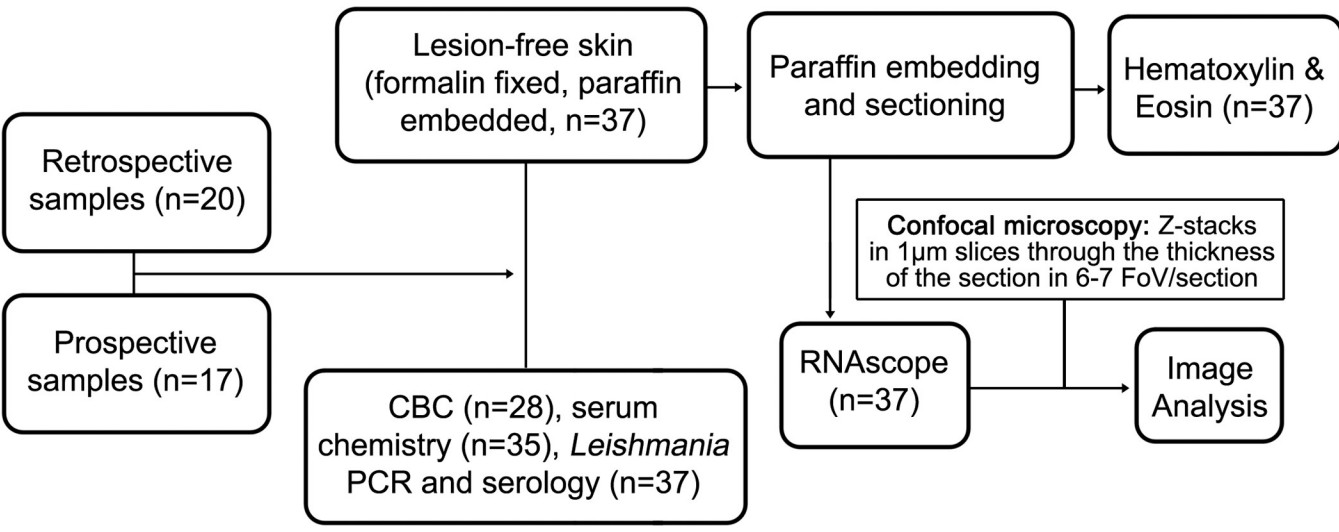

**Fig 1. Study Overview.** Sample collection and processing workflow.

*infantum* is enzootic [24]. All dogs were housed communally in packs and naturally infected with *L. infantum* [26]. LeishVet clinical scores and demographics of dogs are provided (S1 Table).

## Sample collection

For the prospective portion of the study, on the day of sample collection, a veterinarian conducted a physical exam on each dog for clinical signs of leishmaniosis including low body condition score, dermatitis including alopecia, coat condition, lymphadenopathy, splenomegaly, lethargy, conjunctivitis, and pale mucous membranes. Whole blood and serum were submitted for a complete blood count (CBC) and serum chemistry (IDEXX Laboratories Inc.).

Dogs were sedated with dexmedetomidine (Zoetis Inc.) following the manufacterer's published dosing guidelines. Heart rate and respiratory rate were monitored throughout the procedure. Skin biopsies were taken from the cutaneous marginal pouch of the pinna using 3mm punch biopsies (Integra), and subsequent pressure used to achieve hemostasis. The biopsy site was closed with tissue adhesive (3M), and dogs were reversed with atipamezole (Zoetis Inc.) equivalent to the amount of dexmedetomidine used. All dogs recovered well following sedation. In cases of severe clinical disease when requested by the owner, humane euthanasia per AVMA guidelines was performed. For retrospective studies, exams and physicals were performed in a similar manner, with additional details in Scorza et al [24].

## Diagnostics and staging

DNA was isolated from whole blood using the QIAmp Blood DNA Mini Kit according to the manufacturer's instructions (Qiagen). Real Time-quantitative PCR (RT-qPCR) for *Leishmania* ribosomal DNA was performed as previously described [27].

Consistent with current LeishVet guidelines, serology was evaluated by Soluble Leishmania Antigen (SLA) ELISA, a quantitative serological assay [23,28,29]. In some retrospective cases, serology was also evaluated with the Dual-Path Platform® Canine Visceral Leishmaniosis (DPP) serological analysis (ChemBios), detecting antibodies against recombinant *L. infantum* rK28 antigen. Chembios Micro Reader was used for DPP tests to determine quantitative serological status with a reader value > 10 considered as seropositive [30].

To standardize categorization and facilitate discussion of clinical severity, the LeishVet staging system was used to categorize clinical severity based on diagnostic PCR and serology, CBC, Chemistry Panel, and physical exam findings [2]. A higher score indicated more severe clinical disease and more advanced CKD.

## RNAscope and image analysis

Manual RNAscope Multiplex Fluorescent v2 Assay (Advanced Cell Diagnostics) was performed on formalin-fixed, paraffin-embedded canine skin samples, following the manufacterer's protocol, to identify *L. infantum* amastigotes by amastin, and monocytes, a common host cell for *L. infantum*, by CD14 [31]. Briefly, tissue sections were baked for 15 minutes at 60°C and deparaffinized. *Leishmania infantum* amastin (LINF_080012100, S1 Fig) and canine CD14 probes (NCBI Reference Sequence XM_843653.5, S1 Fig), mixed 50:1, were hybridized for 2 hours at 40°C, followed by signal amplification and detection with fluorophores (TSA Vivid Fluorophore 650, Tocris Bioscience and Opal 520, Akoya Biosciences). Slides were counterstained with DAPI and mounted using Prolong Gold (P36930, Life Technologies, Carlsbad, California).

Image analysis is outlined in S2 Fig. Briefly, 6–7 fields of view (FoV) for each slide were imaged with a 40X oil-immersion objective (400X magnification) on a Zeiss LSM 980. FoV

imaged as z-stacks in 1μm slices from 7 to 14 slices, depending on the thickness of the section or region. Z-stacks were flattened and converted to .tiff files using Fiji [32]. In QuPath, the number of amastin$^+$CD14$^+$, amastin$^+$CD14$^-$, amastin$^-$CD14$^+$ cells, and amastin spots/clusters in each image were counted to evaluate the number of infected monocytes, other infected cells, uninfected monocytes, and parasite burden [33].

### Statistical analysis

Comparisons of the estimated mean number of infected monocytes, other infected cells (CD14$^-$), and uninfected monocytes between LeishVet stages were performed using generalized estimating equations (GEE). The estimated mean counts generated by GEE were compared between three groups—Group 1 (LeishVet I), Group 2 (LeishVet II), and Group 3 (LeishVet III & IV)—using the Tukey method. The GEE approach included an exchangeable correlation to account for repeated observations per individual and used robust standard errors. Nonparametric Spearman rank correlations were performed to evaluate correlations between counts, creatinine, and hematocrit because the data was highly skewed. Outliers were identified by ROUT method but kept in the dataset as removal did not change Spearman rank correlations. To further evaluate the relationship between hematocrit and amastin$^+$CD14$^+$ counts, multinomial logistic regression was performed using 3 hematocrit categories as the outcome variable representing moderate/severe anemia, mild anemia, and no anemia: (0,30], (30,37], and (37,60]. Spearman rank correlations were performed using GraphPad PRISM v.10 for Windows (Graphpad Software). All other analysis were performed using R. For all analyses, significance is $^*p \leq 0.05$, $^{**}p < 0.01$, $^{***}p < 0.001$, $^{****}p < 0.0001$.

## Results

### Abundance of infected CD14$^+$ monocytes correlated with dermal parasite burden and infectiousness

The influence of the systemic immune response on parasitemia and clinical disease has been described elsewhere [6]. Here, we evaluated the dermal environment, specifically parasitism of inflammatory CD14$^+$ monocyte cells and other cell types, across LeishVet stages and clinicopathology.

 The demographics of our cohort were consistent with the epidemiology previously seen in this population, with males and adults between 3 and 6 years old over-represented (S1 Table). Consistent with previous microscopy studies of canine leishmaniosis, inflammation was heterogeneously distributed, primarily perivascular and periadnexal in the superficial to mid-dermis [34,35]. Infected monocytes predominately aggregated around hair follicles and glands surrounding hair follicles (Fig 2F–2J). In some LeishVet II and LeishVet III dogs, infected monocytes also began aggregating in the superficial dermis, forming large patches that often coalesced with follicular and glandular inflammatory aggregates (Fig 2K–2O).

 Many samples in the retrospective cohort were collected during a xenodiagnosis study, which quantified dermal parasite burden by PCR and evaluated infectiousness to feeding sand flies of dogs at different stages of clinical disease [24]. Thus, for this subset of individuals (n = 15), we could directly evaluate the relationships between the abundance of infected CD14$^+$ monocytes, as determined here using RNAscope amastin$^+$CD14$^+$ cells, previous estimates of parasite burden derived from qPCR, and infectiousness, measured as the average parasite load per sand fly. We found that the previously estimated dermal parasite burden from qPCR positively correlated with the number of infected CD14$^+$ monocytes (p = 0.0186, r = 0.6079, Fig 3A) and the number of dermal parasites by amastin spots (p = 0.0080,

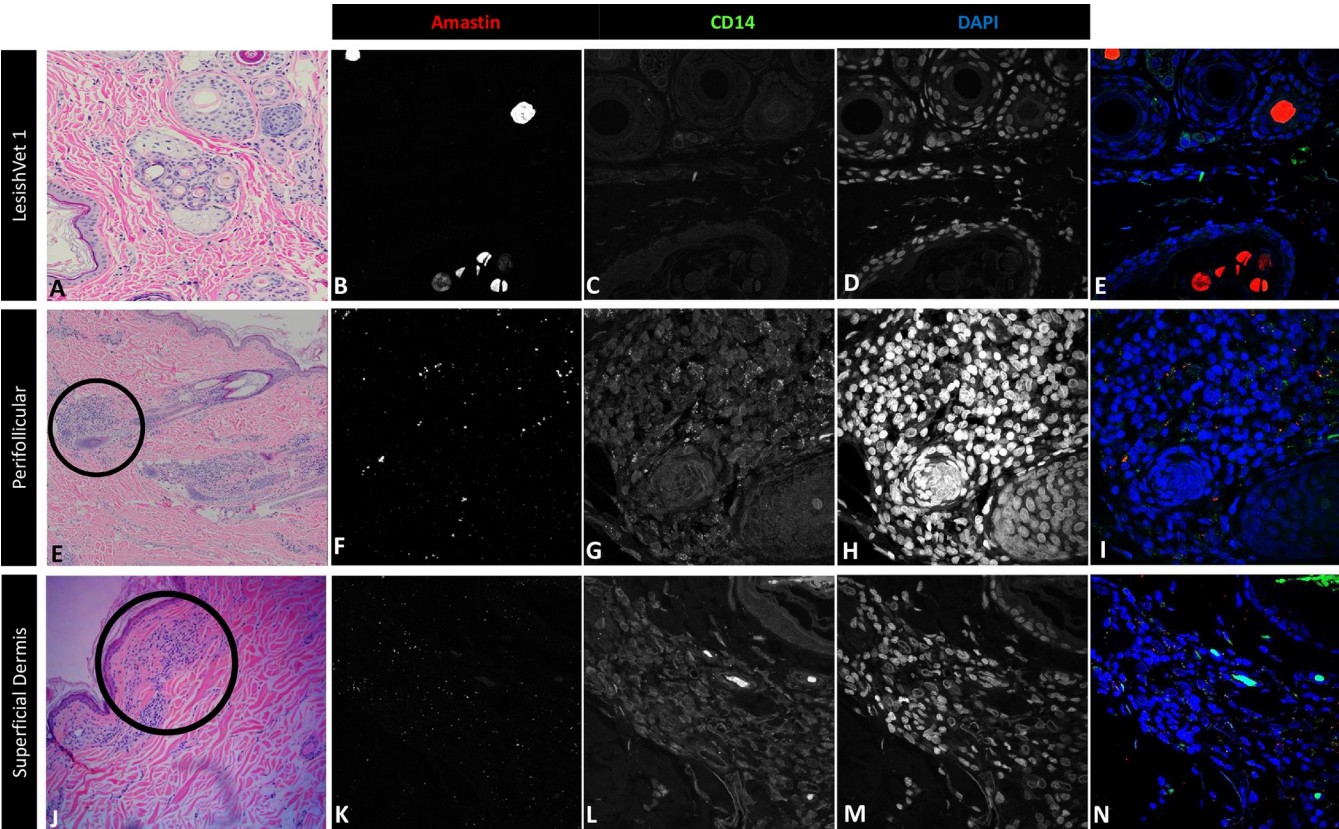

**Fig 2. Amastin+CD14+ cells in the perifollicular area and superficial dermis.** Representative images from a LeishVet I dog with multiple hair follicles and adnexal structures with minimal inflammatory infiltrate (A-E). Aggregates of mixed inflammation surrounding the base of hair follicles in a LeishVet IV dog (E). On RNAscope, CD14+ monocytes (G) infected with *L. infantum* (F) are seen throughout the perifollicular inflammatory aggregate (I). Aggregate of mixed inflammation in the superficial dermis of a LeishVet III dog (J). CD14+ monocytes (L) infected with *L. infantum* (K) dispersed within the superficial dermal inflammatory aggregate on RNAscope (N). Original magnification: 400X (A), 200X (F), 100X (K), 400X (B–E, G–J, L–O).

r = 0.6698, Fig 3B). Individual infectiousness also positively correlated with the number of infected CD14+ monocytes (p = 0.0019, r = 0.7471, Fig 3C), and the number of dermal parasites (p < 0.0001, r = 0.8704, Fig 3D).

## Extent of dermal parasitism varies both between and within LeishVet stages

Descriptive statistics on RNAscope counts for each LeishVet stage are summarized in S2 Table. The results of the GEE analysis showed that LeishVet I dogs (mild disease) were significantly different from dogs in other LeishVet stages, while LeishVet II and III/IV had counts that were not significantly different (Fig 4). The mean number of infected monocytes for LeishVet II dogs was 3.8 times higher than in LeishVet I dogs (18.8 vs 4.8; p = 0.0023). The mean LeishVet III and IV amastin+CD14+ count was 3.6 times higher than the LeishVet I mean count (17.9 vs 4.8; p = 0.0017). Amastin transcripts were also detected in CD14- cells at approximately 2 times higher in LeishVet III and IV dogs (20.6; p = 0.0154) and LeishVet II dogs (18.8; p = 0.0309) compared to LeishVet I dogs (10.2). Additionally, LeishVet I dogs significantly differed from LeishVet III/IV dogs (p < 0.001) and LeishVet II dogs (p < 0.0001) in the number of uninfected CD14+ monocytes. These differences in monocyte counts were not

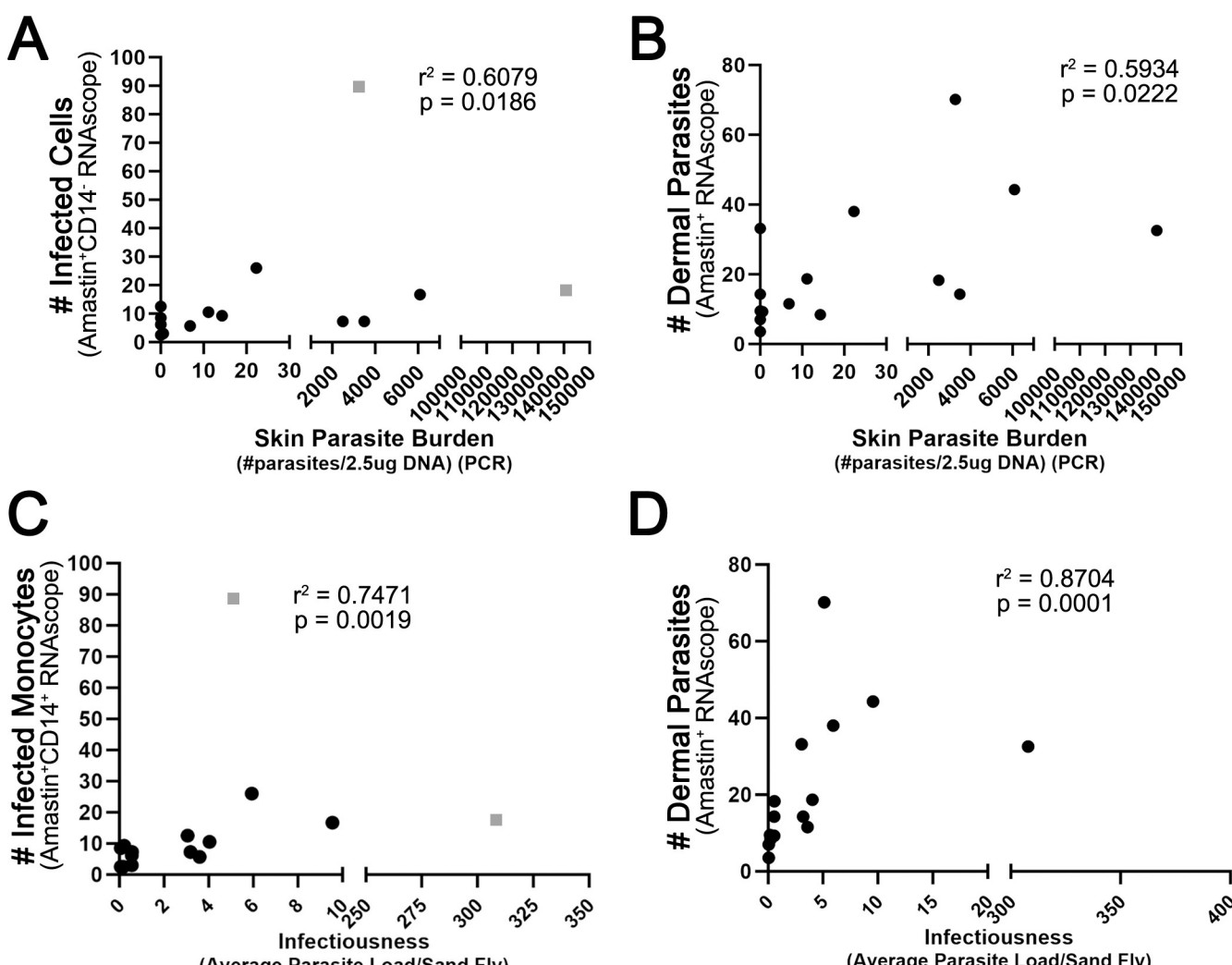

**Fig 3. Increased infected CD14+ monocytes correlated with dermal parasite burden and infectiousness.** Each symbol represents the average infected CD14+ monocyte count for one dog. Average parasite load/sand fly previously calculated and discussed in Scorza, 2021. Statistical outliers indicated with gray squares. Spearman rank correlations remained significant with outliers excluded from analysis.

reflected in the CBC data (n = 27), where aside from 4 dogs, monocyte counts were within the reference range.

While individual counts varied widely, even within the same LeishVet stage, in LeishVet II dogs with mild clinical signs tended to have low counts, similar to LeishVet I, while dogs with moderate clinical signs tended to have higher counts. This was consistent with previous work, where dogs with mild CanL had less inflammation, by semiquantitative histological scoring, and parasite burden, by IHC and qPCR, than dogs with more advanced disease [24,34,35].

### Abundance of infected CD14+ monocytes inversely correlated with kidney function measured by serum creatinine

Chronic kidney disease (CKD) due to progressive renal damage is commonly described in CanL, and the LeishVet classification system uses the International Renal Interest Society (IRIS) CKD staging guidelines to differentiate clinical severity [2,7,8]. LeishVet I and II dogs

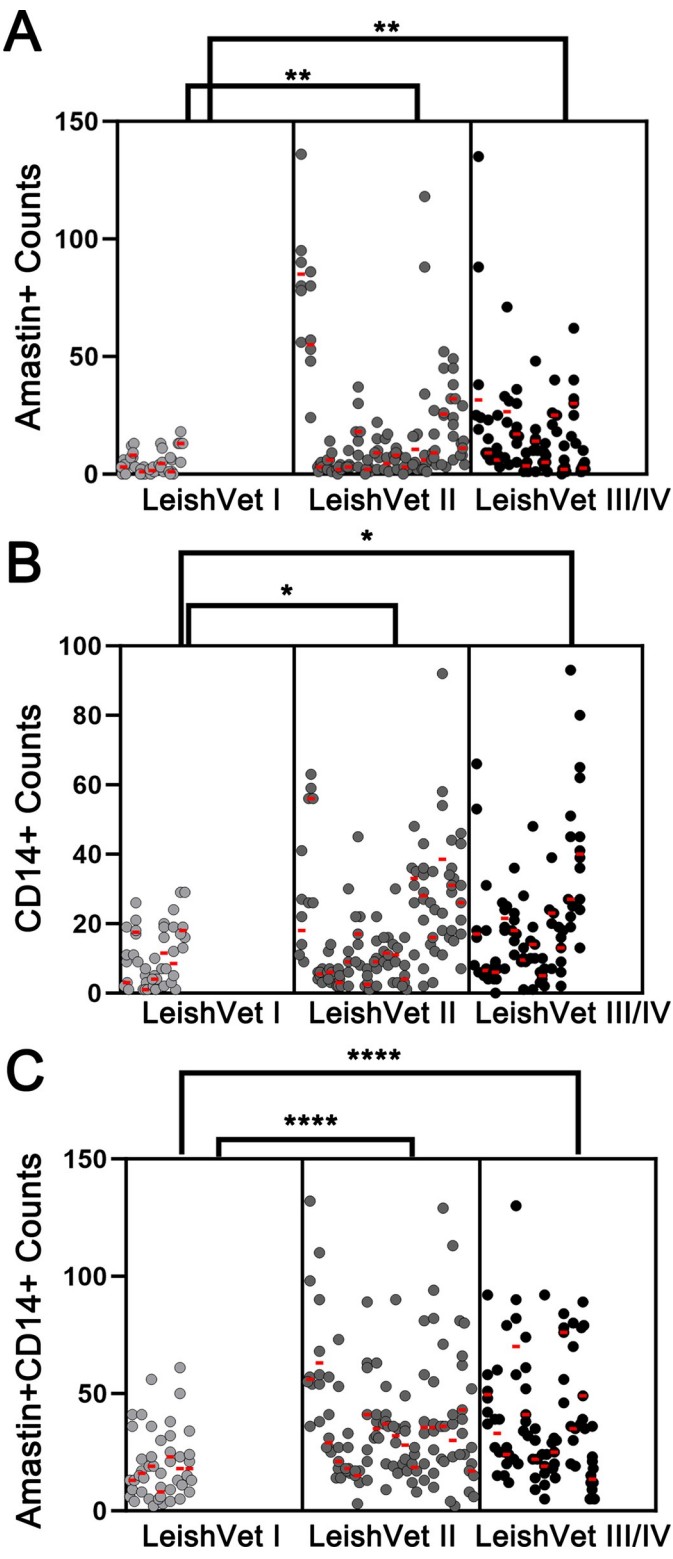

**Fig 4. LeishVet I dogs have fewer infected dermal monocytes than dogs at later disease stages.** Counts of infected CD14$^+$ monocytes (A), infected CD14$^-$ cells (B), and uninfected CD14$^+$ monocytes (C) across LeishVet stages. Each column of symbols is one dog. Each symbol represents the count for one 40X field. The red line highlights the median count for each individual. Generalized estimating equations.

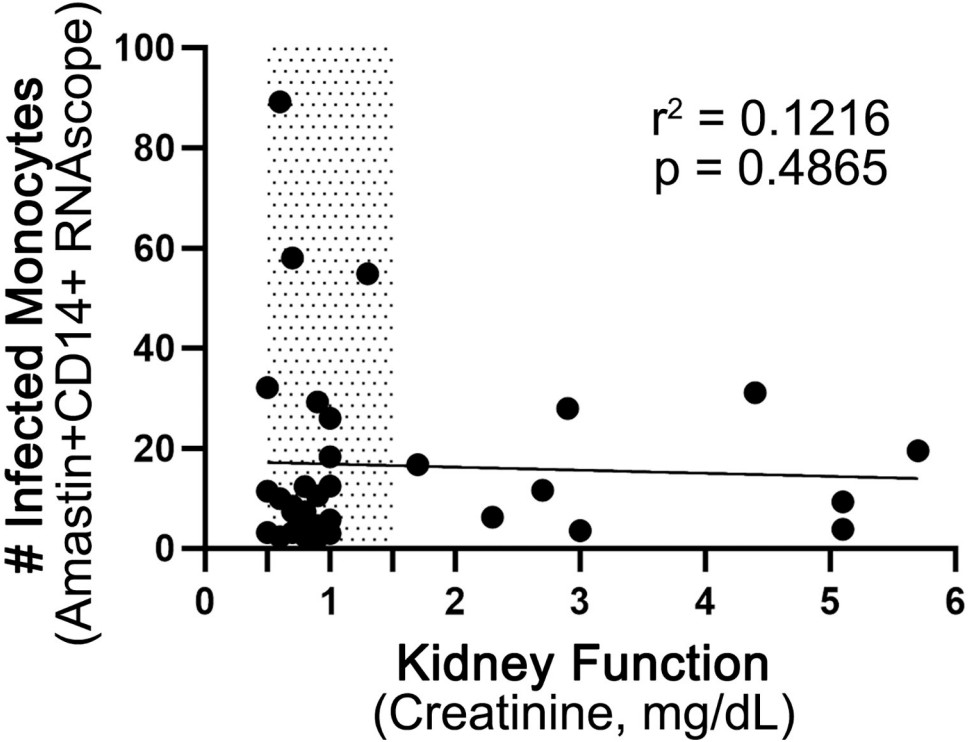

**Fig 5. Relationship between dermal monocyte parasitism and serum creatinine.** Average infected CD14[+] monocyte counts for all dogs with the reference range for creatinine shaded. Spearman rank correlation.

have IRIS Stage 1 CKD, while IRIS Stage 2 CKD classifies a dog as LeishVet III [2,36]. Progressive CKD causes widespread physiological derangement due to electrolyte imbalances and accumulation of toxic waste products, such as urea, in the blood [37]. As uremia has been shown to impair systemic immune cell function, including monocytes [37–41], we examined the relationship between serum creatinine and the number of infected CD14[+] monocytes in the skin. While not statistically significant, there was a trend suggesting an inverse correlation between infected CD14[+] monocytes and serum creatinine, with cell counts decreasing 1.19-fold as creatinine increased 4.5-fold between IRIS Stage 1 CKD and IRIS Stage 2–4 CKD (Fig 5).

### Anemia significantly correlates with increased numbers of infected CD14[+] monocytes

Moderate to severe cases of CanL frequently present with a mild to moderate non-regenerative anemia [2]. Based on the relationship between bone marrow and hematologic/erythroid abnormalities in CanL, we investigated the relationship between the number of infected dermal CD14[+] monocytes and hematocrit (HCT), an indicator of anemia on routinely run bloodwork. Infected CD14[+] monocyte numbers were negatively correlated with HCT (p = 0.0098, r = -0.4798, Fig 6A [42]). The average number of infected monocytes was 2.33 fold higher in dogs with mild anemia (HCT 30–37%) compared to dogs with no anemia (HCT > 37%), and 3.23 fold higher in dogs with moderate to severe anemia (HCT <30%) compared to dogs with no anemia [42].

Because the renal pathology in progressive CKD can also cause anemia, we repeated the analysis, focusing only on LeishVet II dogs. Abundance of infected CD14[+] monocytes

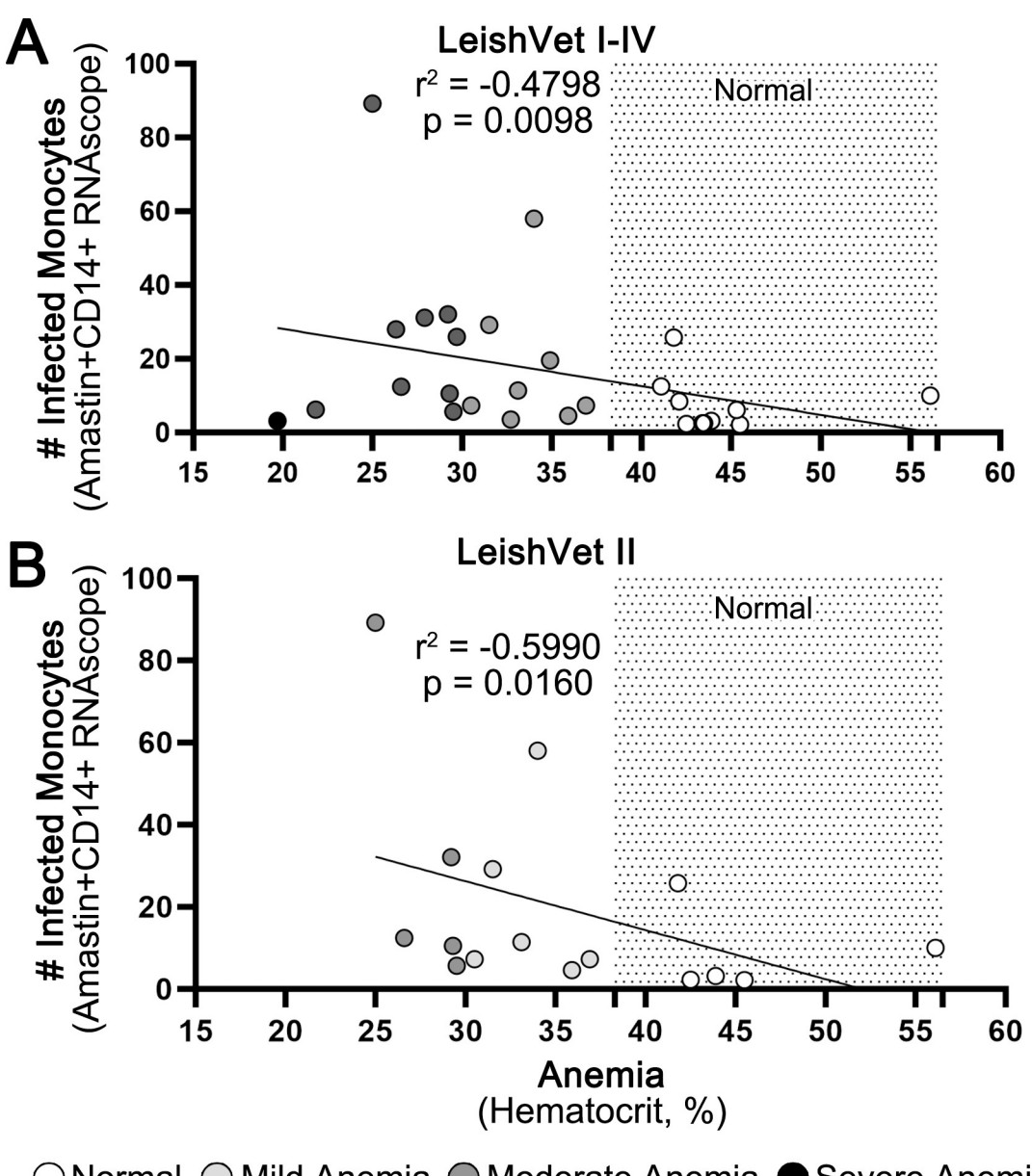

**Fig 6. Increased dermal parasitism correlates with anemia.** Average infected CD14[+] monocyte count for all dogs (A) and for LeishVet II dogs (B) with the hematocrit reference range shaded. Each symbol is one dog. Light grey symbols are dogs with mild anemia (hematocrit 30–37%). Dark grey symbols are dogs with moderate anemia (hematocrit 20–29%). Black symbols are dogs with severe anemia (hematocrit < 20%). Spearman rank correlation.

remained correlated with decreasing hematocrit (p = 0.0393, r = -0.5416, Fig 6B). LeishVet II dogs also had similar fold changes in infected CD14[+] monocytes–a 2.25-fold increase between dogs with no anemia vs mild anemia, and a 3.45 fold increase between dogs with no anemia vs moderate/severe anemia.

Multinomial logistic regression analysis did not identify factors significantly influencing HCT values. However, when predicted probabilities of HCT based on infected CD14[+] monocyte counts were calculated, no anemia (HCT > 37%) had the lowest probability, which consistently decreased as the average number of infected CD14[+] monocytes increased.

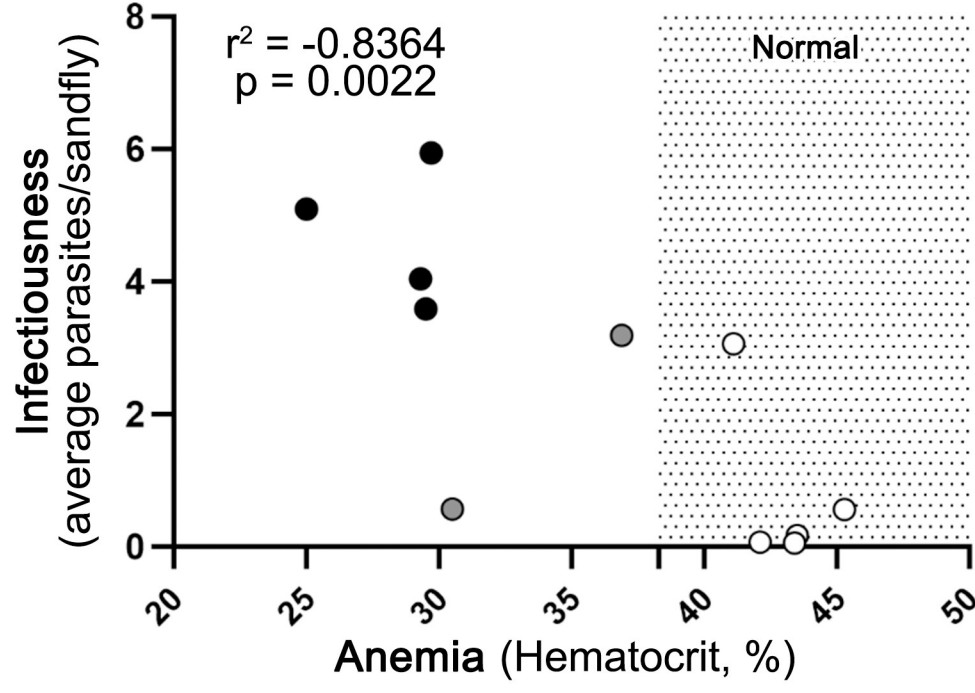

**Fig 7. Anemia was associated with increased infectiousness.** Each symbol represents the average value for one dog, and the reference range for hematocrit is shaded. Average parasite load/sand fly previously calculated and discussed in Scorza, 2021. Spearman rank correlations.

### Anemia was associated with greater infectiousness

Based on the significant correlations between dermal infected and uninfected CD14⁺ monocyte counts and skin parasite burden, infectiousness to sand flies, and hematocrit, we wanted to determine if hematocrit correlated with infectiousness. Within skin samples obtained after xenodiagnosis, a lower hematocrit (i.e. anemia) correlated with increased average parasite load/sand fly (p = 0.0022, r = -0.8364, Fig 7). The average parasites per sand fly increased almost 6-fold between dogs with no anemia vs moderate or severe anemia and increased approximately 2.5-fold between dogs with no anemia vs mild anemia (2.40) and between mild vs moderate or severe anemia (2.48).

### Discussion

This study builds upon previous findings on the relationships between dermal parasitism, inflammation distribution, clinical disease, and transmissibility [24,34,35,43]. Consistent with this previous work, we found that dermal parasitism and inflammation significantly increased once dogs developed clinical signs. Dogs with a low hematocrit (an indicator of anemia) had significantly more infected CD14⁺ monocytes in the skin and were more infectious to feeding sand flies than dogs without anemia. Importantly, we did not find any significant associations with increasing creatinine (an indicator of chronic kidney disease). Our data suggests that parasitism of key hematopoietic organs and resultant anemia influences infectiousness to sand flies by affecting the systemic and dermal immune environments and dermal parasitism.

In addition to infected CD14+ monocytes, CD14- cells were identified on RNAscope. These cells could represent an uncommon host cell type, such as a fibroblast, or a subtype of

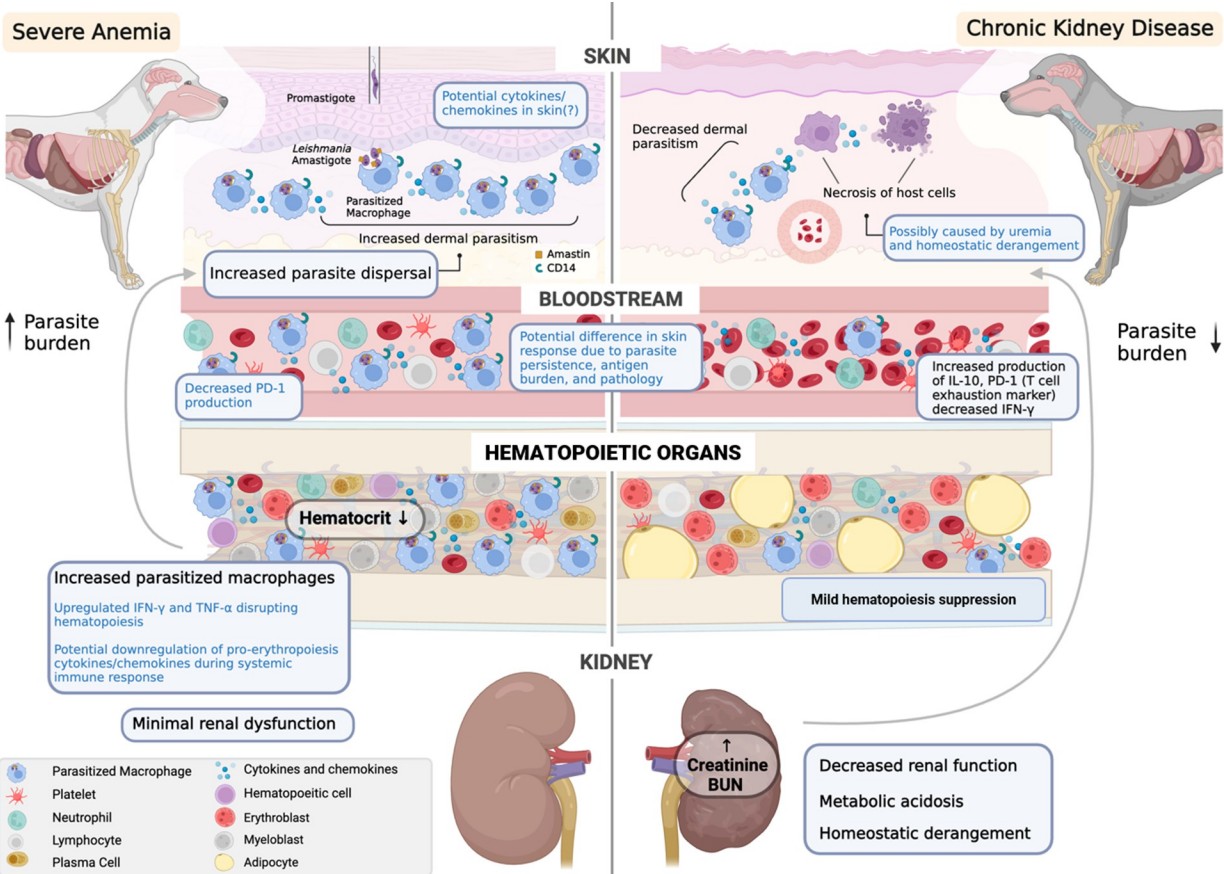

**Fig 8. Hypothesized mechanisms for the effects of hematopoietic and renal pathology on dermal parasitism in canine leishmaniosis.** Proposed mechanisms listed in blue text. Created by Soomin Koh using BioRender.

phagocytic cell that does not express CD14, such as some populations of canine dendritic cells and monocytes [44–46]. Additional analysis of the transcriptional profile and protein expression of these cells will enable more accurate identification and evaluation of their role in dermal parasitism and infectiousness to sand fly vectors.

Because renal failure and hematologic abnormalities in CanL result from different pathologic mechanisms, we hypothesize that this affects how they influence the dermal environment, dermal parasitism, and–ultimately—infectiousness (Fig 8). Advanced CKD causes uremia, acid-base imbalances, and other physiologic derangements that create a systemically inhospitable environment [37]. As part of the generalized immunosuppression, chronic uremia causes monocytes to become more pro-inflammatory but less able to phagocytize or present antigen [41]. In addition to impairment of monocytes, the common host cell for *L. infantum*, chronic uremia also affects the skin, where parasite transmission occurs, through damage to the dermal vasculature [37,47]. Consequently, a harsher dermal environment in LeishVet III and IV dogs with advanced, chronic CKD may adversely affects both infiltrating immune cells and any *Leishmania* amastigotes they harbor, decreasing parasite viability and host infectiousness (Fig 8).

Multiple pathologic mechanisms in CanL can contribute to anemia [12,13], but we hypothesize that bone marrow pathology plays a significant role, as it is the primary hematopoietic organ. Infiltration of parasitized cells in bone marrow crowds out progenitor cells and shift the

cytokine environment by favoring production of inflammatory cytokines, such as TNF and IFNγ, that inhibit erythropoiesis, which can result in anemia [13–15,48–50]. In addition to inhibiting erythropoiesis and creating anemia, bone marrow parasitism could promote the accumulation of infected and uninfected CD14[+] monocytes in the skin by facilitating entry of parasitized cells into circulation and dispersal to peripheral tissues (Fig 8). However, other pathologic mechanisms, such as increased hemophagocytosis, or organs, such as the spleen, could be driving the relationship between anemia, dermal parasitism, and infectiousness [12,13].

This work builds on previous research illustrating the role of dogs with mild to moderate clinical CanL to the epidemiology of *Leishmania*. Current public health interventions often do not target this population, focusing primarily on dogs with severe clinical disease. Further education of veterinarians and public health officials about this population, particularly the correlation between infectiousness, dermal parasite burden, and anemia, would help increase awareness and inform interventions. Additionally, further research into the mechanisms linking anemia, dermal parasitism, and infectiousness is needed to identify effective treatment strategies.

## Supporting information

**S1 Fig. Probe Sequences.** Leishmania infantum amastin and canine CD14 RNAscope probes.
(DOCX)

**S2 Fig. Image analysis pathway.** Z-stacks from confocal microscope (1) converted to flax maximum intensity projections in Fiji (2). In Fiji, projection split into C-1, C-2, and C-3 (3). Brightness and contrast were optimized for each channel, and the channels were merged back together. In QuPath, the area of interest in each image was annotated, excluding non-inflammatory structures (4). Cells were counted within the annotation (5), and then separate object classifiers for amastin[+] (6) and CD14[+] (7) based on maximum signal intensity in a cell. A composite classifier was then created to detect double positive cells (8). Intracellular amastin spots (max size 2μm$^2$) and clusters counted using QuPath's subcellular spot detection feature to determine the number of dermal parasites (8).
(TIF)

**S1 Table. Cohort Demographics.** Summary of the age in years, sex, LeishVet status, and 4DX SNAP results for tickborne diseases of all dogs. Multiple* includes 2 animals positive for *Borrelia burgdorferi/Ehrlichia* and 1 animal positive for *Borrelia burgdorferi/Anaplasma*.
(DOCX)

**S2 Table. Descriptive Statistics for RNAscope Counts.** N indicates the number of 40X fields analyzed.
(DOCX)

## Acknowledgments

The authors would like to acknowledge use of the University of Iowa Central Microscopy Research Facility, a core resource supported by the University of Iowa Vice President for Research, and the Carver College of Medicine. The CMRF's acquisition of the Zeiss LSM 980 microscope used in this research was made possible by a generous grant from the Roy J. Carver Charitable Trust, with additional funding provided by the University's Office of the Vice President for Research. The authors would like to thank the University of Iowa Comparative Pathology Laboratory technicians for their invaluable help with sample embedding and

sectioning; as well as the animal caretakers for their longstanding collaboration in support of this work.

## Author Contributions

**Conceptualization:** Max C. Waugh, Paul M. Kaye, Christine A. Petersen.

**Data curation:** Max C. Waugh.

**Formal analysis:** Max C. Waugh, Ferney Henao-Ceballos, Jacob J. Oleson.

**Funding acquisition:** Paul M. Kaye, Christine A. Petersen.

**Investigation:** Max C. Waugh, Karen I. Cyndari, Tom J. Lynch, Christine A. Petersen.

**Methodology:** Max C. Waugh, Tom J. Lynch, Paul M. Kaye.

**Project administration:** Max C. Waugh, Christine A. Petersen.

**Resources:** Paul M. Kaye, Christine A. Petersen.

**Supervision:** Christine A. Petersen.

**Visualization:** Max C. Waugh, Karen I. Cyndari, Tom J. Lynch, Soomin Koh.

**Writing – original draft:** Max C. Waugh.

**Writing – review & editing:** Karen I. Cyndari, Tom J. Lynch, Ferney Henao-Ceballos, Jacob J. Oleson, Christine A. Petersen.

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
