## [Decision Letter · Decision Letter 0]

12 Aug 2024

Dear Dr. Waugh,

Thank you very much for submitting your manuscript "Clinical anemia predicts dermal parasitism and reservoir infectiousness during progressive visceral leishmaniasis" for consideration at PLOS Neglected Tropical Diseases. As with all papers reviewed by the journal, your manuscript was reviewed by members of the editorial board and by several independent reviewers. In light of the reviews (below this email), we would like to invite the resubmission of a significantly-revised version that takes into account the reviewers' comments. 

We cannot make any decision about publication until we have seen the revised manuscript and your response to the reviewers' comments. Your revised manuscript is also likely to be sent to reviewers for further evaluation.

Sincerely,

Johan Van Weyenbergh

Academic Editor

Laura-Isobel McCall

Section Editor

Reviewer's Responses to Questions

**Key Review Criteria Required for Acceptance?**

**Methods**

-Are the objectives of the study clearly articulated with a clear testable hypothesis stated?

-Is the study design appropriate to address the stated objectives?

-Is the population clearly described and appropriate for the hypothesis being tested?

-Is the sample size sufficient to ensure adequate power to address the hypothesis being tested?

-Were correct statistical analysis used to support conclusions?

-Are there concerns about ethical or regulatory requirements being met?

Reviewer #1: Discussion on the tick coinfection status of these dogs is missing, particularly with respect to the known Ehrlichia endemicity in this cohort and its ability to induce anemia. Whether Ehrlichia serology was performed at the time of skin collection for the cohort should be described and the proportion of positive dogs within this cohort. For the anemia correlations, separating by Ehrlichia+ and Ehrlichia- dogs would be suggested, where Ehrlichia- dog anemia can more strongly be attributed to Leishmania induced bone marrow alterations.

Reviewer #2: The methods section is detailed and thorough. The use of RNAscope and confocal microscopy is appropriate for evaluating dermal parasitism. The explanation of the LeishVet clinical stages and the parameters measured (anemia, creatinine) is clear. The inclusion of a CONSORT flow diagram for sample collection is a good practice, enhancing transparency.

Reviewer #3: The methods were appropriate for the study described. In the section starting at line 153 the researchers should restate what the outcome is for the GEE model more specifically than "mean counts." It is not clear if multiple GEE models were run for each of the following outcomes "number of infected monocytes, other infected cells, 

uninfected monocytes, and parasite burden" (stated in lines 150 and 151) or just one model. The researchers should also consider including a supplemental document showing the different LeishVet stages in more detail. The authors reference the clinical stages but a table describing the staging may be more helpful for readers.

Reviewer #4: First, I would like to congratulate the authors on the study. It presents methodological rigor, presents defined objectives, has a reasonable study design, complies with ethical requirements and performs correct statistical analysis. Finally, minor modifications should be made to bring more clarity to the study.

These will be listed below:

1. Onychogryphosis was not mentioned, as this sign/symptom is pathognomonic of canine visceral leishmaniasis. Was this symptom evaluated in the animals observed? If not, justify why it was not observed.

2. What dosage of dexmedetomidine was used? Was the animal's weight assessed? And what was the animal's reaction after anesthesia? I think it is worth detailing the entire anesthetic protocol.

3. The authors used the ELISA and DPP techniques, in addition to RT-qPCR. The ELISA technique is not the gold standard for antigen-antibody evaluation. I recommend including RIFI or justifying why it is not used.

**Results**

-Does the analysis presented match the analysis plan?

-Are the results clearly and completely presented?

-Are the figures (Tables, Images) of sufficient quality for clarity?

Reviewer #1: Figures S3 and S4 are very similar, and I do not see S4 referenced anywhere in the text, so one should be removed. S4 format is preferred showing the individual channels and should be moved to the main text. This figure should also include a representative image set from a LeishVet stage I dog for comparison.

Figure 2 legend, line 197, outliers indicated with X. No X appears in the figure, update. 

Figure 4-6, the focus on analysis just of infected monocytes leaves out important information you have access to with this data. Clearly amastin is found in CD14- cells, in fact you have higher amastin+CD14- counts than amastin+CD14+ counts in every group, indicating another host cell exists. Therefore total amastin or CD14-amastin+ correlations should also be performed and commented on against all relevant targets. Either add figure panels or state in the text, expand the results to include the correlation results between creatinine and anemia against the other combinations of amastin: total amastin counts (amastin+CD14-/+) or amastin+CD14-.

Reviewer #2: - The results are well-presented, with clear distinctions made between the different stages of disease and their corresponding levels of dermal parasitism and infectiousness. The significant correlations between anemia and infectiousness are convincingly shown. The lack of correlation with creatinine reinforces the study’s focus on anemia. The discussion aptly interprets the results, emphasizing the importance of anemia in predicting infectiousness. The authors provide a logical argument for why classical measures of disease severity, like renal failure, do not correlate with infectiousness. The implications for public health initiatives are well-articulated, suggesting a shift in focus to early signs of disease.

- All figures and tables are clearly labeled and referenced in the text

Reviewer #3: The analyses presented match the analysis plan. The authors indicate in the section "Anemia significantly correlates with increased numbers of infected CD14+ monocytes" at line 237 how the distribution of dogs with anemia relates to the LeishVet scoring.

Reviewer #4: The analysis presented corresponds to the analysis plan, with clear and complete results. I recommend evaluating the resolution of Figure 7, as it presents areas of quality blur.

**Conclusions**

-Are the conclusions supported by the data presented?

-Are the limitations of analysis clearly described?

-Do the authors discuss how these data can be helpful to advance our understanding of the topic under study?

-Is public health relevance addressed?

Reviewer #1: The discussion of CKD and urea’s potential effects on the immune response and local skin microenvironment are nicely discussed, although an association between CKD and dermal parasite RNA was not found. A significant association was found between anemia and dermal parasite RNA and infectiousness to sand flies, yet the discussion between how anemia could influence the dermal immune environment or sand fly feeding is lacking. Please add some discussion of mechanisms that may link systemic anemia to skin parasitism or increased uptake by sand flies. Relatedly, is anemia correlated with blood or spleen parasite burden in this cohort? That could support the systemic load is higher in dogs with anemia.

Reviewer #2: The conclusion succinctly summarizes the main findings and their public health implications. The recommendation to focus on anemic dogs for breaking the zoonotic cycle is well-supported by the data

Reviewer #3: No comments. The conclusions are supported by the data presented.

Reviewer #4: They are within the requested standards.

**Editorial and Data Presentation Modifications?**

Reviewer #1: Line 36, the authors statement that anemia influenced infectiousness to sand flies is overinterpreting. A correlation was observed but may be independent. No testing a direct link between anemia and infectiousness to sand flies was performed. Please use correlated to be more appropriate. 

Lines 90-91, the conclusion of that study was not that moderate diseased dogs are more infectious than severely diseased dogs, but that infectiousness to sand flies was most correlated with dermal parasite burden over clinical severity. 

Figures S3 and S4 are very similar, and I do not see S4 referenced anywhere in the text, so one should be removed. S4 format is preferred showing the individual channels and should be moved to the main text. This figure should also include a representative image set from a LeishVet stage I dog for comparison.

Figure 2 legend, line 197, outliers indicated with X. No X appears in the figure, update.

Reviewer #2: (No Response)

Reviewer #3: Minor edits: 

At line 22 consider changing "were considered" to "have previously been considered."

At line 31 the authors should change LeishVet 2 to LeishVet II. Roman numerals are used throughout the rest of the manuscript.

At line 56 please change "influences" to "influence."

Starting at line 89 in the introduction the authors should consider expanding on the LeishVet clinical scoring system. The authors can then elaborate further on the importance of using the LeishVet scores in the methods and highlight how the scoring system provides a more robust understanding of clinical stage and in concordance infectiousness in the discussion section.

At line 122 considering changing text to read "with additional details in Scorza et al."

The authors should consider adding additional information in the discussion to highlight the public health impact of developing targeted interventions to reduce disease transmission in dogs with less severe disease. The authors touch on that importance in the introduction but mainly highlight the reasoning behind the higher parasite burden in the discussion. Several sentences on the public health impact of this study would be valuable at the end of the discussion.

Reviewer #4: The study has potential and is a great contribution to the field of leishmaniasis. I recommend that, in addition to the methodology suggestions, the epidemiological aspect be included. For example, the analysis of the clinical characteristics collected was not categorically analyzed. In addition, data collection on associated factors should be included, such as: whether the animal frequently visits the veterinarian, uses any skin care products, the climate/temperature of the place of residence, among others. I am sure that the addition of these analyses will bring more robustness to the study.

**Summary and General Comments**

Reviewer #1: Canine dermal skin parasitism by Leishmania is thought to be a strong determinant of host infectiousness to sand fly vectors, which spread the parasite to humans. The authors utilize a cutting edge technique, RNAscope, to probe a cohort of retrospective and prospective CanL skin sections to compare dermal Leishmania Amastin RNA and canine CD14 RNA counts between dogs with a range of clinical severity based on the LeishVet staging system. This method performed well on the canine skin samples and demonstrates the strong usefulness of this tool for expanding the ability of the field to explore in situ dermal infection and the skin immune microenvironment in canines. Interesting findings include CD14+ cells are not the only host cell in the skin and that anemia may influence skin parasite load and transmissibility to sand fly vectors. The mechanisms linking anemia to increased skin parasite load, whether a direct mechanism affecting accumulation of parasites in the skin or that anemia is simply a biomarker for increased bone marrow infection and therefore systemic parasite burden, are not fully fleshed out.

Reviewer #2: The manuscript provides an insightful exploration into the relationship between clinical anemia and dermal parasitism in dogs infected with Leishmania infantum, and its implications for infectiousness to sand flies. The study's focus on anemia as a predictor of infectiousness in dogs with L. infantum offers a novel perspective and addresses an important gap in understanding the progression of leishmaniosis that could influence future public health policies targeting zoonotic visceral leishmaniasis (VL). The manuscript is well-written, methodologically sound, and contributes meaningfully to the field, it is suitable for publication in PLOS Neglected Tropical Diseases

Reviewer #3: Overall the manuscript was well written and addresses an important aspect of canine leishmaniosis, infectiousness as it relates to disease progression. The study highlights the high infectiousness of dogs with mild to moderate disease, a population not primarily targeted by most current public health interventions. The work underscores our need to better understand not just disease progression but also transmission.

Reviewer #4: (No Response)

PLOS authors have the option to publish the peer review history of their article (what does this mean?). If published, this will include your full peer review and any attached files.

Reviewer #1: No

Reviewer #2: No

Reviewer #3: No

Reviewer #4: No
---

## [Decision Letter · Decision Letter 1]

28 Oct 2024

Dear Dr. Waugh,

We are pleased to inform you that your manuscript 'Clinical anemia predicts dermal parasitism and reservoir infectiousness during progressive visceral leishmaniosis' has been provisionally accepted for publication in PLOS Neglected Tropical Diseases.

Best regards,

Johan Van Weyenbergh

Academic Editor

Laura-Isobel McCall

Section Editor

Shaden Kamhawi

co-Editor-in-Chief

Paul Brindley

co-Editor-in-Chief

Reviewer's Responses to Questions

**Key Review Criteria Required for Acceptance?**

**Methods**

-Are the objectives of the study clearly articulated with a clear testable hypothesis stated?

-Is the study design appropriate to address the stated objectives?

-Is the population clearly described and appropriate for the hypothesis being tested?

-Is the sample size sufficient to ensure adequate power to address the hypothesis being tested?

-Were correct statistical analysis used to support conclusions?

-Are there concerns about ethical or regulatory requirements being met?

Reviewer #1: (No Response)

Reviewer #4: All review questions have been answered or addressed. I recommend accepting.

**Results**

-Does the analysis presented match the analysis plan?

-Are the results clearly and completely presented?

-Are the figures (Tables, Images) of sufficient quality for clarity?

Reviewer #1: (No Response)

Reviewer #4: All review questions have been answered or addressed. I recommend accepting.

**Conclusions**

-Are the conclusions supported by the data presented?

-Are the limitations of analysis clearly described?

-Do the authors discuss how these data can be helpful to advance our understanding of the topic under study?

-Is public health relevance addressed?

Reviewer #1: (No Response)

Reviewer #4: All review questions have been answered or addressed. I recommend accepting.

**Editorial and Data Presentation Modifications?**

Reviewer #1: (No Response)

Reviewer #4: (No Response)

**Summary and General Comments**

Reviewer #1: (No Response)

Reviewer #4: All review questions have been answered or addressed. I recommend accepting.

PLOS authors have the option to publish the peer review history of their article (what does this mean?). If published, this will include your full peer review and any attached files.

Reviewer #1: No

Reviewer #4: No

---

## [Editor Report · Acceptance letter]

3 Nov 2024

Dear Dr. Waugh,

We are delighted to inform you that your manuscript, "Clinical anemia predicts dermal parasitism and reservoir infectiousness during progressive visceral leishmaniosis," has been formally accepted for publication in PLOS Neglected Tropical Diseases.

Best regards,

Shaden Kamhawi

co-Editor-in-Chief

Paul Brindley

co-Editor-in-Chief
